# Effects of Music Therapy on Vital Signs in Children with Chronic Disease

**DOI:** 10.3390/ijerph19116544

**Published:** 2022-05-27

**Authors:** Susann Kobus, Alexandra M. Buehne, Simone Kathemann, Anja K. Buescher, Elke Lainka

**Affiliations:** 1Center of Artistic Therapy, University Medicine Essen, 45147 Essen, Germany; 2Clinic for Pediatrics II, University Children’s Hospital Essen, 45147 Essen, Germany; alexandra.buehne@web.de (A.M.B.); simone.kathemann@uk-essen.de (S.K.); anja.buescher@uk-essen.de (A.K.B.); elke.lainka@uk-essen.de (E.L.)

**Keywords:** music therapy, pediatrics, vital signs, chronic diseases, intensive care unit, ICU, special care unit, SCU

## Abstract

Background: Recent research found evidence supporting music therapy for hospitalized children with chronic diseases. The aim of this study was to investigate the effect of music therapy on hospitalized children’s vital signs. Methods: In this prospective study, children with chronic gastroenterological and nephrological diseases received active or receptive music therapy two to four times a week until discharge from hospital at the pediatric special care unit (SCU) and pediatric intensive care unit (ICU). Baseline and post-therapy heart rate, oxygen saturation and blood pressure were recorded and analyzed as control values at three points on the same day when the children were alone in their patient room at rest. Results: A total of 83 children, median 3 age of years (range one month to eighteen years) received music therapy. In total, 377 music therapy sessions were treated: 200 receptive therapy (78 ICU, 122 SCU) and 177 with active therapy (0 ICU, 177 SCU). Music therapy interventions showed changes in vital signs during music therapy sessions. After music therapy, heart rates decreased by 18 beats per minute (95% confidence interval (CI), −19.4 to (−16.8)), oxygen saturation increased by 2.3% (95% CI, 2.2 to 2.5), systolic blood pressure decreased by 9.2 (95% CI, −10.6 to −7.7) and diastolic blood pressure decreased by 7.9 (95% CI, −9.6 to −6.3). When music therapy was applied at the SCU (ICU), heart rates significantly reduced by 17.9 (18.9) beats per min, oxygen saturation increased by 2.4% (2.1%) and blood pressure reduced by 9.2 (2.8) mmHg (systolic) and 7.9 (0.3) mmHg (diastolic). Almost all control values were better than directly before the intervention. However, after music therapy intervention, the children showed better values in vital signs compared to being alone in their patient room. Conclusion: Music therapy is an added value for children with kidney and liver/gastrointestinal diseases during their hospital stay.

## 1. Introduction

The diagnosis of a chronic disease in children is principally associated with physical and mental strain in patients of all age groups and their families [1].

In particular, children undergoing liver or kidney transplantation are hospitalized for a longer period of time after transplantation. During this period, due to decreased social, physical activities and high infection risks, the patients generally develop intense emotional responses which include loneliness, depression, and rejection, finally leading to anger, anxiety, and confusion [2]. Depression, anxiety, or stress can lead to an increase in heart rate [3]. As a result, blood flow to the heart decreases and the body produces more cortisol. Over time, these effects can lead to heart disease. In recent decades, non-pharmacological approaches such as music therapy and art therapy have been implemented to a certain extent in the treatment of pediatric patients worldwide [4]. Music therapy is becoming a standard supportive care service in many pediatric hospitals across the United States. More than half of the respondents in a study reported serving in high acuity areas such as the pediatric intensive care, hematology/oncology, or neonatal intensive care units [5]. Music therapy has become a consolidated strategy to relieve stress in children during hospitalization, and previous research demonstrated its efficacy on individuals’ health [6]. Improvisational musical activity with therapeutic objectives and outcomes has been found to facilitate motivation, communication skills and social interaction, as well as sustaining and developing attention [7]. In general, patients, due to illness or treatment effects, are physically weak and lack interest in performing any activities that involve physical movement. Therefore, among all non-pharmacological approaches, music therapy is considered as one of the most favorable choices for patients, as it does not involve big physical movement [8,9].

A study with cognitive behavioral therapy showed increases in heart rate variability in patients with Major Depression. Initially more depressed patients showed the most pronounced cardiovascular improvements through cognitive behavioral therapy. These findings provide insights into the effects of psychological treatment against depression [10]. However, previous studies with music therapy have shown that music therapy has a stabilizing and relaxing effect on both the infant’s general behavioral status and physiological parameters such as oxygen saturation, heart rate and respiratory rate [11,12,13,14,15].

The aim of this study was to examine the effects of music therapy on hospitalized children’s vital signs, depending on the type of medical care (intensive care vs. general pediatric special care) and the kind of music therapy. We hypothesized that performing music therapy on children and adolescents up to the age of eighteen has an impact on the vital signs heart rate, oxygen saturation and blood pressure, regardless of the strength of the diseases.

## 2. Methods

### 2.1. Study Design

The study was designed as a prospective clinical trial. Depending on their state of health, the children were treated at a pediatric intensive care unit (ICU) or pediatric special care unit (SCU) and received active or receptive music therapy. In receptive music therapy, the children listen to the music played individually by the music therapist. In active music therapy, the patient has the option of improvising with their own voice, with instruments alone, or together with the music therapist. Apart from music therapy, there were no differences in medical care. To investigate the effects of music therapy on vital signs of heart rate, oxygen saturation and the blood pressure, we analyzed the protocols of the music therapy sessions. To compare these values, we analyzed the same vital signs, heart rate, oxygen saturation, and the blood pressure, three times on the same day of the music therapy sessions, when the children stood in their room without any other treatment. All parameters were referenced to the clinic’s internal normal values. Oxygen saturation was measured using a pulse oximeter clip on the patients’ fingers. In babies up to one year, oxygen saturation was measured on the heel.

### 2.2. Inclusion Criteria and Exclusion Criteria

All children in pediatric care at the University Hospital Essen on the SCU and the ICU between November 2020 and October 2021 up to the age of eighteen, with chronic gastroenterological and nephrological diseases and a state of health with the possibility for conducting music therapy, were eligible for the study. The reasons for exclusion were insufficient German language skills to understand the objectives of the study, a lack of interest to participate in the study and no parents for consent. Further exclusion criteria were difficulties in organizing the time of therapy due to a large number of clinical examinations and instable patients who received music therapy in palliative care.

Parental and child consent forms were obtained. The study was approved by the local ethics committee of the Medical Faculty of the University of Duisburg-Essen (19-9003-BO) and registered with DRKS DRKS00026158.

### 2.3. Intervention

Music therapy was performed two to four times a week in clinically stable patients from admission to the clinic until discharge. The timing of each therapy session was coordinated by the music therapist, patients, nursing staff and parents. Each music therapy session with the children was carried out in the own patients’ room directly next to the bed and consisted of individual conducted music therapy by the music therapist. The instruments used during the therapy were disinfected by the music therapist before and after entering the patient rooms, according to standardized hygiene concepts. Various instruments, including several percussion instruments and therapeutic instruments such as sansula, ocean disc and zaphir, xylophones, tone bars, egg-shakers, vibraslaps, cajons, bongos, a keyboard and a guitar, were available for the therapy. The therapist brought all these instruments to the children’s room and their use depended on the patients’ choice during the therapy. If the children were very weak or tired and not able to play actively, music therapy treatment was changed into receptive music therapy with the instrument sansula played by the music therapist.

The vital signs, such as heart rate, oxygen saturation and blood pressure, of the patients were documented following protocol before and after each therapy session. Clinical data were retrieved from the electronic patient files and the intensive care files.

### 2.4. Statistical Analysis

Quantitative variables are presented as mean and 95% confidence intervals or standard deviation. Univariate statistical analyses were used for clinical parameters. The Shapiro–Wilk test was used to determine the distribution of continuous variables (mean and standard deviation for normal distribution, median and range for non-normal distribution). Therapy sessions were stratified depending on the age at the intervention into age groups 0 to 1 year, 1 to 3 years, 4 to 6 years, 7 to 10 years, 11 to 14 years and 15 to 18 years. Therapy sessions were classified depending on the kind of care they were conducted into ICU and SCU. Depending on the kind of music therapy, we separated interventions into the two categories active and receptive music therapy.

To compare the values of vital signs before and after music therapy, paired *t*-tests were performed. All statistical calculations were performed using IBM SPSS Statistics 27 (IBM, Chicago, IL, USA). *p*-values < 0.05 were considered significant.

## 3. Results

### 3.1. Patients

Eighty-three children (male 48, female 35) were included in the study. The included children had a median age of three years (range one month to eighteen years) at the first therapy session. Thirty patients were less than one year old. Table 1 shows the clinical characteristics of the included patients.

### 3.2. Music Therapy Sessions

A total of 377 music therapy sessions were conducted until the age of eighteen years (Table 2). The median duration of each music therapy session was 41 ± 1.3 min (range 12 and 70 min). We found an overall decrease in heart rate after music therapy of 18.1 beats per min (95% CI, −19.4 to −16.8) in all patients. Stratification by age showed that the decrease was present in all age groups (Table 2). We saw an increase of 2.4% (95% CI, 2.2 to 2.5) for all children when comparing values of oxygen saturation before and after music therapy. Stratification showed that the increase was also present in all age groups (Table 2). When analyzing blood pressures, we saw a decrease in systolic blood by 9.2 mmHg and in diastolic blood by 7.9 mmHg. The decreases were also present in all age groups (Table 2).

In all age groups, we saw lower control values in heart rates at all three times on the same day of the music therapy session. However, except for eleven to fourteen-year-old children, all control values in heart rates correspond to the standard values (Table 3). The oxygen saturation at the control times was almost always higher than before a music therapy session, but lower after a music therapy intervention. Only in the eleven to fourteen-year-old children, the oxygen saturation was higher before the music therapy session than at only two control times, and in the one to three-year-old children, the oxygen saturation was lower after the music therapy than at the control measurements. All control values are similar to the clinic’s internal standard values. Immediately before the music therapy, the systolic blood pressure was usually higher than at the control measurement times, except in the four to six-year-old children. There was one control value higher than before music therapy. After music therapy, the systolic blood pressure value was always lower than before music therapy and at control measurements, except in those children with an age under one year. The control values for the one to three-year-old children and the four to six-year-old children are rather high compared to the standard values. The diastolic blood pressure values were similar to the systolic measurements. The control values for the under one year old children are high compared to the standard values. All pre-music therapy values were higher than control measurements, except for the four to six-year-old children. There is one control value higher than before the music therapy (Table 4). All pre- and post-music therapy values and control values in heart rates are higher in the hospitalized children with chronic diseases than in healthy children (Table 3). The systolic and diastolic blood pressure increases with higher age in healthy children (Table 3) but increases and decreases irregularly regardless of age in pre- and post-music therapy values and control values (Table 2 and Table 4).

### 3.3. Music Therapy at ICU vs. SCU

A total of 299 (79%) sessions were conducted at the SCUs and 78 (21%) at the ICU. When analyzing the vital signs, dependent of the kind of clinical unit, we saw a decreased heart rate and an increased oxygen saturation in both types of units (Table 5). We found an overall decrease in heart rate of 17.9 beats per min (95% CI, −19.3 to −16.5) in music therapy interventions at the SCU and of 18.9 beats per min (95% CI, −22.0 to −15.8) in music therapy interventions at the ICU. We saw an increase of 2.4% (95% CI, 2.2 to 2.5) for all children at the SCU when comparing values of oxygen saturation before and after music therapy. At the ICU, we found an increase of 2.1% (95% CI, 1.6 to 2.5) (Table 5). Blood pressures decreased at the SCUs significantly (systolic 9.4 mmHg and diastolic 7.4 mmHg) and at the ICU not significantly (systolic 2.8 mmHg and diastolic 0.3 mmHg) (Table 5).

At the ICU, the control heart rate values of 127.1, 129.2 and 129.5 were lower than immediately before a music therapy intervention, but significantly higher than after music therapy. Additionally, at the SCU, we found lower control values (111.9, 120.9, 113.9) of heart rates compared to before the music therapy intervention, but higher than after the intervention. We also saw higher control oxygen saturation values than before music therapy at the SCU (98.3, 98.5, 98.6) and at two control values at the ICU (98.4, 96.3, 98.1), but lower values at both types of units than after music therapy sessions. The control systolic blood pressures at the SCU (108.5, 109.6, 110.2) were lower than before music therapy und higher after music therapy, but at the ICU, the control values were lower than before and after music therapy sessions (87.2, 86.2, 88.1). The diastolic blood pressures at the SCU (65.7, 66.3, 66.9) were lower than before music therapy und higher after music therapy, but at the ICU, the control values were higher than before and after music therapy sessions (55.2, 54.6, 54.9). The blood pressure is lower at the ICU than at the SCU and is not influenced by music therapy.

### 3.4. Active vs. Receptive Music Therapy

Out of 377 sessions, 200 (53%) were performed as receptive music therapy and 177 (47%) as active music therapy. The median duration of each active music therapy session was 47 ± 1.6 min (range 20 and 60 min) and of each receptive music therapy session was 35 ± 1.6 min (range 12 and 70 min). At the ICU, we only had receptive music therapy sessions (Table 6). At the SCU, we conducted 177 active and 122 receptive music therapy sessions (Table 6).

We found an overall decrease in heart rate at the SCU (16.6 beats per min active, 19.6 beats per min receptive) and at the ICU (18.9 beats per min receptive) after music therapy. Both values of receptive music therapy, baseline and post-therapy were higher at the ICU than at the SCU (Table 6). We saw an overall increase in oxygen saturation at the SCU (2.2% active, 2.7% receptive) and at the ICU (2.1% receptive therapy) after music therapy. Both baseline values of oxygen saturations in receptive music therapy at the SCU and ICU were the same, but post-therapy values at the SCU were higher than at the ICU (Table 6).

We found a decrease in systolic blood pressure at the SCU after active and receptive music therapy, but we saw a non-significant decrease in systolic blood pressure at the ICU after receptive therapy (Table 6). In diastolic blood pressure at the SCU, we also saw a decrease after active and receptive music therapy, and we also saw a non-significant decrease in diastolic blood pressure at the ICU after receptive therapy (Table 6). All values of receptive music therapy, baseline and post-therapy were lower at the ICU than at a SCU. At the ICU, we had no active music therapy (Table 6).

At the ICU, we had only receptive music therapy. Control values in heart rates (128.8, 132.2, 132.8) were lower than directly before a music therapy session, but higher than after an intervention. The control values in oxygen saturation (98.2, 96.6, 97.8) were higher than directly before the intervention, but lower than after a music therapy session. The control values in systolic (83.9, 83.6, 84.1) and diastolic (54.5, 54.0, 54.1) blood pressures were higher than before and after the music therapy.

At the SCU, receptive and active music therapy was conducted for the children. At receptive music therapy, we saw lower control heart rates (123.8, 131.6, 119.3) than directly before music therapy and higher rates after the intervention. When compared with active music therapy (control rates 109.8, 113.2, 113.1), we found the same trend. The same relationship we saw in oxygen saturation (receptive: 98.9, 98.1, 98.8; active: 98.2, 98.4, 98.3).

At active music therapy, control values in systolic blood (112.1, 116.1, 118.2) and diastolic blood 68.1, 69.6, 72.1) were higher before and lower after music therapy.

At receptive music therapy, control values in systolic blood (104.9, 99.4, 101.6) and diastolic blood (62.9, 60.2, 59.8) were lower before and also lower after music therapy.

## 4. Discussion

Music therapy is an efficient method of therapy to complement medical treatment. The evidence shows that music therapy should be prescribed routinely for patients in the ICU [16]. According to a study, a single music therapy intervention can be effective over the session period in intubated patients during the weaning phase [17]. Our study confirms evidence that music therapy has beneficial effects in hospitalized children. In our prospective intervention study with live music therapy, we could also show that music therapy stabilizes vital signs in hospitalized children with chronic gastroenterological and nephrological diseases, reflected by decreased heart rates of 18.1 beats per min (95% CI, 19.4 to 16.8) and increased oxygen saturation of 2.4% (95% CI, 2.2 to 2.5) after sessions. Our results agree with the findings in preterm infants that music therapy has a stabilizing and relaxing effect on vital functions, which is manifested in a decrease in heart rate and an increase in oxygen saturation [11]. All pre- and post-music therapy values and control values in heart rates are higher in the hospitalized children with chronic diseases than in healthy children, which could be due to the fact that chronically ill nephrological children often have blood pressure problems with arterial hypertension and take medication that also affects heart rate [18]. The changes in heart rate measured in our study are very different to the increases in heart rate variability in patients with Major Depression [10].

Additionally, concerning the effects on heart rates and oxygen saturation in children up to eighteen years with chronic gastroenterological and nephrological diseases at active and receptive music therapy sessions at the SCU and ICU, we found that there is only a slight difference in the effectiveness of active or receptive music therapy. Receptive and active music therapy had equal effects in children. Heart rates and oxygen saturation improved during both kinds of music therapy. In terms of heart rate and oxygen saturation, patients benefitted equally from the music therapy intervention, regardless of the severity of the disease. The heart rate (−17.9 beats per min SCU, −18.9 beats per min ICU) and oxygen saturation (2.4% SCU, 2.1% ICU) improved nearly independently, whether the child was at the SCU or at the ICU. The changes in heart rate and oxygen saturation values are almost the same. However, baseline and post-therapy values of heart rates were lower and of oxygen saturation were higher in children at the SCU than at the ICU, which can be explained by the severity of the disease [19]. Blood pressures decreased at the SCUs significantly (systolic 9.4 mmHg and diastolic 7.4 mmHg). However, the changes in systolic (−2.8 mmHg) and diastolic blood pressure (0.3 mmHg) at the ICU have no clinical significance. This shows that the positive effects of music therapy of blood pressure decrease with increasing disease severity. All children were not sedated at the time of music therapy. In a future study, the mental state of the children should also be analyzed in addition to the vital signs before and after a music therapy session. When comparing active and receptive music therapy at the SCU, we found that receptive music therapy is more efficient. Stratification of the units showed that at the ICU, we only used receptive music therapy, which can be explained by the severity of the illness and the intensive medical care associated with it. We did not see any significant changes in blood pressure values during the receptive music therapy at the ICU. Music therapy did not affect blood pressure. These findings are in line with a study with hypertensive patients [20].

When analyzing the control measurements in nearly all age groups, we saw that the heart rates and blood pressures immediately before a music therapy intervention are higher than at the control measurements, which can be attributed to the fact that the children initially become tense when an intervention is imminent. During the music therapy session, a feeling of well-being arises and the heart rate drops again and is significantly below the control values, which shows that music therapy is beneficial for the children. Heart rate fluctuations can also occur at any time. With increasing age, heartbeat variability decreases, which can cause the heart rate to drop rapidly. Body temperature can also affect heart rate [21]. With fever, the heart rate may increase, and when it is cold, the heart rate may decrease. Fluctuations in blood pressure can arise with kidney disease, increasing age and obesity. Sleep lowers blood pressure [22]. Strong fluctuations and high blood pressure can be dangerous [23]. The oxygen saturation is lower immediately before music therapy, but significantly higher after music therapy than at control measurements, which also shows that music therapy relaxes the children. Normal fluctuations in oxygen saturation can be caused by the pH value and body temperature increase. Due to respiratory diseases and by centralizing the circulation, the oxygen saturation decreases. Oxygen saturation is falsely low due to cold extremities or hypothermia [24].

Our results confirm that live music by the bedside is an additional, simple and inexpensive factor in the ICU to transform the critical care setting into a more familiar and domestic environment [25].

Our results are in line with the findings that the integration of music therapy in inpatient treatment of adolescents is feasible and acceptable and is valued by staff and patients as a complement to “talking therapies”. Participation is enjoyed and associated with outcomes including improvement in mood, expression of feelings and social engagement consistent with recovery [26].

Several studies have shown a significant reduction in psychological distress and an increase in well-being through music therapy. Music interventions in people with cancer can have positive effects on anxiety, pain, mood, and quality of life, but have little impact on heart rate, respiratory rate, and blood pressure [27]. Music therapy, in pediatric oncology, seems to have a good feasibility and positive effects on mental and physical health [6]. There are also positive effects of music therapy with female patients with breast cancer. The patients who received music therapy stayed 13.62 (±2.04) days shorter in hospital than the patients who did not receive music therapy [28]. Music therapy techniques, especially receptive methods, are feasible and well accepted in terminally ill cancer patients. Therapeutic conversation seems to play an important role. Frequency and duration of music therapy, previous experience with music and music therapy, as well as sociodemographic factors influence the positive effects of music therapy [29]. Parents also perceived positive outcomes for their children who received music therapy [30].

We found no study that investigated music therapy not in the field of pediatric oncology, but in the context of hospitalized children with chronic gastroenterological and nephrological diseases.

Our study has some limitations. There was no comparison to the vital signs of a control group who received no music therapy intervention or received a different type of therapy. Could music play an important role in the care of critically ill patients? A future study should investigate whether music therapy shows greater benefits and can be more effective than other forms of support. The limitations of our analysis include varying quality of data documentation, heterogeneity of patients and different disease groups. Interaction with co-existing diseases and co-medication could not be analyzed in detail. The heart rate and blood pressure are age dependent. Our results do not differentiate whether the heart rate and blood pressure were increased or normal before the music therapy session.

## 5. Conclusions

In conclusion, our study confirms former findings on the stabilizing effect of live music therapy on the vital signs of heart rate, oxygen saturation and blood pressure. It adds new evidence that music therapy is effective for hospitalized children with chronic gastroenterological and nephrological diseases, regardless of how well the children are doing. Music therapy is also very effective for seriously ill children at the ICU but does not show any stabilization of blood pressure.

## Figures and Tables

**Table 1 ijerph-19-06544-t001:** Clinical characteristics of participants.

	Male	Female	All Patients
Patients, n (%)	48 (58)	35 (42)	83 (100)
gastroenterological patients	29 (55)	24 (45)	53 (100)
nephrological patients	19 (63)	11 (37)	30 (100)
Number of music therapy sessions, n (%)	273 (72)	104 (28)	377 (100)
Median duration of music therapy sessions, min. (range)	40 (15–60)	42 (12–70)	41 (12–70)
Clinical units of music therapy sessions, n (%)	87 (68)	41 (32)	128 (100)
Gastroenterological unit	108 (77)	32 (23)	140 (100)
Nephrological unit	21 (68)	10 (32)	31 (100)
Preterm infants with gastroenterological and nephrological illnesses intensive care unit	57 (73)	21 (27)	78 (100)
Median age, years at music therapy	3	3	3

**Table 2 ijerph-19-06544-t002:** Heart rate, oxygen saturation and blood pressure before and after music therapy sessions.

Vital Signs	Age,	Sessions,	Mean before Therapy	Mean after Therapy	Mean Difference
	Years	n	(95% CI)	(95% CI)	(95% CI)
Heart rate/min	all	377	131.1 (128.4–133.3)	113.0 (111.0–115.0)	−18.1 (−19.4–(−16.8))
	<1	158	144.1 (141.4–146.7)	125.2 (122.7–127.8)	−18.8 (−20.8–(−16.8))
	1–3	81	129.1 (123.9–134.3)	109.5 (105.2–134.3)	−19.6 (−22.9–(−16.3))
	4–6	70	122.3 (118.00–126.57)	104.0 (100.07–107.88)	−18.3 (−21.0–(−15.6))
	7–10	17	114.7 (107.2–122.1)	101.7 (94.9–108.4)	−13.0 (−16.1–(−8.9))
	11–14	19	115.1 (109.3–120.9)	101.7 (96.3–107.2)	−13.4 (−16.5–(−10.2))
	15–18	32	103.3 (96.5–110.0)	87.7 (81.8–93.6)	−15.5 (−19.6–(−11.5))
SaO_2,_ %	all	377	96.9 (96.7–97.1)	99.2 (99.1–99.4)	2.4 (2.2–2.5)
	<1	158	96.7 (96.3–97.1)	99.0 (98.8–99.3)	2.4 (2.1–2.7)
	1–3	81	93.1 (89.0–97.3)	95.7 (91.5–99.9)	2.6 (2.2–2.9)
	4–6	70	97.1 (96.7–97.6)	99.6 (99.4–99.8)	2.5 (2.0–2.9)
	7–10	17	97.2 (96.3–98.2)	99.1 (98.5–99.7)	1.9 (0.8–2.8)
	11–14	19	97.5 (96.8–98.3)	99.4 (99.0–99.8)	1.9 (1.2–2.6)
	15–18	32	96.9 (96.4–97.4)	99.0 (98.5–99.5)	2.1 (1.5–2.6)
Blood, systolic, mmHg	all	210	116.4 (114.5–118.3)	107.2 (105.5–109.0)	−9.2 (−10.6–(−7.7))
	<1	51	108.0 (103.0–113.1)	101.0 (96.6–105.4)	−7.0 (−10.6–(−3.5))
	1–3	53	120.2 (117.4–123.1)	108.4 (105.4–111.3)	−11.9 (−15.2–(−8.6))
	4–6	59	121.1 (118.3–124.0)	111.1 (108.5–113.8)	−10.0 (−11.9–(−8.0))
	7–10	15	114.6 (106.9–122.3)	104.7 (96.9–112.4)	−9.9 (−13.4–(−6.5))
	11–14	18	112.6 (106.6–118.6)	105.4 (99.0–111.8)	−7.2 (−12.1–(−2.3))
	15–18	31	116.3 (12.5–120.1)	107.4 (103.1–111.8)	−8.9 (−12.7–(−5.1))
Blood, diastolic, mmHg	All	210	71.5 (70.1–72.9)	35.7 (32.4–38.9)	−7.9 (−9.6–(−6.3))
	<1	51	68.1 (64.3–72.0)	62.8 (60.5–65.0)	−5.3 (−8.7–(−3.2))
	1–3	53	73.7 (70.7–76.8)	64.5 (62.5–66.4)	−9.3 (−12.4–(−6.1))
	4–6	59	73.9 (72.0–75.8)	65.2 (63.4–66.9)	−8.7 (−10.1–(−7.4))
	7–10	15	69.9 (67.2–72.6)	62.0 (59.3–64.8)	−7.9 (−10.3–(−5.6))
	11–14	18	71.2 (68.0–74.3)	66.2 (62.8–69.6)	−4.9 (−8.4–(−1.5))
	15–18	31	72.4 (69.4–75.3)	67.0 (64.3–69.6)	−5.4 (−8.6–(−2.2))

**Table 3 ijerph-19-06544-t003:** Standard values in heart rates and systolic and diastolic blood pressure.

Vital Signs	Age, Years	Standard Values without Music Therapy
Heart rate at rest/min	0.5	110–175
	1–3	80–140
	4–6	75–130
	7–10	70–120
	11–13	60–100
Blood, systolic, mmHg	0.5	80–110
	1–3	80–113
	4–6	80–115
	7–10	83–122
	11–13	95–136
	14–16	100–127
Blood, diastolic, mmHg	0.5	43–63
	1–3	46–79
	4–6	47–79
	7–10	52–83
	11–13	58–88
	14–16	55–77

**Table 4 ijerph-19-06544-t004:** Heart rate, oxygen saturation and blood pressure at three control measurements without music therapy.

Vital Signs	Age, Years	First Mean	Second Mean	Third Mean
		Control Measurement	Control Measurement	Control Measurement
Heart rate/min	<1	138.2	132.1	131
	1–3	116.5	115.8	114.8
	4–6	117.5	113.7	112.1
	7–10	104.2	112.3	104.9
	11–14	106	107.9	100.2
	15–18	96	95.8	96
SaO_2,_ %	<1	98.7	97.4	98.3
	1–3	96.1	97.6	96.6
	4–6	98.1	98.5	99.4
	7–10	98.4	97.8	98.8
	11–14	97.6	97.3	96.7
	15–18	97.9	97.7	97.5
Blood, systolic, mmHg	<1	91.3	99.8	99.5
	1–3	113	111.8	114.4
	4–6	114.6	116.2	121.7
	7–10	106.2	108	107.5
	11–14	108.7	108.7	109.5
	15–18	112.3	115.4	112.1
Blood, diastolic, mmHg	<1	65.6	65.4	64.1
	1–3	68.1	68.4	69.6
	4–6	70.8	70.8	75.8
	7–10	65.4	67.6	63.8
	11–14	68.8	68.6	68.4
	15–18	69.1	68.9	68.9

**Table 5 ijerph-19-06544-t005:** Heart rate, oxygen saturation and blood pressure before and after music therapy sessions at the SCU and ICU.

Vital Signs	Unit	Sessions, n	Mean before Therapy (95% CI)	Mean after Therapy (95% CI)	Mean Difference (95% CI)	*p*-Value
Heart rate/min	SCU	299	128.2 (125.9–130.6)	110.4 (108.3–112.4)	−17.9 (−19.3–(−16.5))	0.000
	ICU	78	141.5 (136.0–147.0)	122.6 (117.2–127.9)	−18.9 (−22.0–(−15.8))	0.000
SaO_2_, %	SCU	299	97.0 (96.8–97.2)	99.4 (99.3–99.6)	2.4 (2.2–2.6)	0.000
	ICU	78	96.5 (95.9–97.0)	98.6 (98.1–99.0)	2.1 (1.6–2.5)	0.000
Blood, systolic, mmHg	SCU	199	117.3 (115.6–119.0)	107.9 (106.3–109.5)	−9.4 (−10.8–(−7.9))	0.000
	ICU	8	94.3 (71.1–117.5)	91.5 (70.4–112.6)	−2.8 (−14.8–9.3)	0.605
Blood, diastolic, mmHg	SCU	199	72.3 (70.9–73.6)	64.9 (63.9–65.9)	−7.4 (−8.7–(−6.0))	0.000
	ICU	8	53.8 (44.4–63.1)	53.5 (46.9–60.1)	−0.3 (−4.9–4.4)	0.903

**Table 6 ijerph-19-06544-t006:** Heart rate, oxygen saturation and blood pressure before and after active and receptive music therapy sessions at the SCU and ICU.

Vital Signs	Kind of Therapy	Unit	Sessions, n	Mean before Therapy (95% CI)	Mean after Therapy (95% CI)	Mean Difference (95% CI)	*p*-Value
Heart rate/min	Active	SCU	177	124.3 (121.5–127.1)	107.7 (105.1–110.3)	−16.6 (−18.5–(−14.8))	0.000
		ICU	0	--	--	--	--
	Receptive	SCU	122	134.0 (130.4–137.8)	114.4 (111.3–117.5)	−19.6 (−21.7–(−17.5))	0.000
		ICU	78	141.5 (136.0–147.0)	122.6 (117.2–127.9)	−18.9 (−22.0–(−15.8))	0.000
SaO_2_, %	Active	SCU	177	97.3 (97.1–97.6)	99.5 (99.4–99.7)	2.2 (1.9–2.4)	0.000
		ICU	0	--	--	--	--
	Receptive	SCU	122	96.5 (96.1–97.0)	99.2 (99.0–99.5)	2.7 (2.4–3.1)	0.000
		ICU	78	96.5 (95.9–97.0)	98.6 (98.1–99.0)	2.1 (1.6–2.5)	0.000
Blood, systolic, mmHg	Active	SCU	157	119.2 (117.5–120.9)	109.9 (108.2–111.6)	−9.3 (−11.0–(−7.6))	0.000
		ICU	0	--	--	--	--
	Receptive	SCU	42	110.3 (105.8–114.9)	111.5 (96.7–104.4)	−9.8 (−12.8–(−6–7))	0.000
		ICU	8	94.3 (71.0–117.5)	91.5 (70.4–112.6)	−2.8 (−14.8–9.3)	0.605
Blood, diastolic, mmHg	Active	SCU	157	72.9 (71.6–74.3)	65.4 (64.4–66.5)	−7.5 (−8.9–(−6.1))	0.000
		ICU	0	--	--	--	--
	Receptive	SCU	42	69.8 (65.8–73.8)	63.0 (60.3–65.7)	−6.8 (−10.4–(−3.1))	0.001
		ICU	8	53.8 (44.4–63.1)	53.5 (46.9–60.1)	−0.3 (−4.9–4.4)	0.903

## Data Availability

Original data will be made available to any qualified researcher upon request.

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
