# Peer review of "Effects of Music Therapy on Vital Signs in Children with Chronic Disease"

_ijerph, 2022, doi:10.3390/ijerph19116544_

Round 1
Reviewer 1 Report
Dear author team, many thanks for giving me the opportunity to review your manuscript. Research in 2014 shows that music interventions may have beneficial effects on anxiety, pain, mood, and quality of life in adults with cancer and may have a small effect on heart rate, respiratory rate, and blood pressure (https://doi.org/10.1155/2014/103297). Very important is the distinction between active and received music therapy, which in my opinion sometimes is overlooked or not explained in depth. However, you point out different types of MT right away. It is unfortunate that there was no other control group (no intervention). This could have turned this important piece of research into a controlled clinical trial. The researchers could also discuss the placebo effect in their discussion (an intervention as opposed to no intervention usually creates significant differences in a before/after design.
Author Response
Dear Reviewer,
we highly appreciate your critical input and the opportunity to improve our manuscript entitled " Effects of music therapy on vital signs in children with chronic disease". Our changes to the original manuscript are highlighted in red.
All comments were addressed in a point-to-point fashion below:
COMMENT 1
Dear author team, many thanks for giving me the opportunity to review your manuscript. Research in 2014 shows that music interventions may have beneficial effects on anxiety, pain, mood, and quality of life in adults with cancer and may have a small effect on heart rate, respiratory rate, and blood pressure (https://doi.org/10.1155/2014/103297).
We included the following text to the manuscript:
Music interventions in people with cancer can have positive effects on anxiety, pain, mood, and quality of life, but have little impact on heart rate, respiratory rate, and blood pressure [25].
COMMENT 2
Very important is the distinction between active and received music therapy, which in my opinion sometimes is overlooked or not explained in depth. However, you point out different types of MT right away. It is unfortunate that there was no other control group (no intervention). This could have turned this important piece of research into a controlled clinical trial. The researchers could also discuss the placebo effect in their discussion (an intervention as opposed to no intervention usually creates significant differences in a before/after design.
We included the following text to the manuscript:
Our study has some limitations. There was no comparison to the vital signs of a control group that got no music therapy intervention or received a different type of therapy. A future study should investigate whether music therapy shows greater benefits and can be more effective than other forms of support.
Reviewer 2 Report
My main concern is that the work establishes that hospitalization can lead patients to a state of depression and/or anxiety, but it is not something that they have measured. I believe that the introduction should be reconsidered so that it is established that the expected improvement is in physiological parameters.
It would be convenient to explain why these parameters were chosen and how they influence the improvement of the treatment for which they are hospitalized and the other associated comorbidities.
There is a difference in the gravity of the patient depending on whether their problem is gastric or renal. Do they present more significant physical wear or decay?
How convenient would it be to apply a scale to any age group to determine their state of mind? Thus, the rationale presented by the authors and the physiological part measured could be correlated.
Author Response
Dear Reviewer,
we highly appreciate your critical input and the opportunity to improve our manuscript entitled " Effects of music therapy on vital signs in children with chronic disease". Our changes to the original manuscript are highlighted in red.
Our comment is addressed below:
My main concern is that the work establishes that hospitalization can lead patients to a state of depression and/or anxiety, but it is not something that they have measured. I believe that the introduction should be reconsidered so that it is established that the expected improvement is in physiological parameters.
It would be convenient to explain why these parameters were chosen and how they influence the improvement of the treatment for which they are hospitalized and the other associated comorbidities.
There is a difference in the gravity of the patient depending on whether their problem is gastric or renal. Do they present more significant physical wear or decay?
How convenient would it be to apply a scale to any age group to determine their state of mind? Thus, the rationale presented by the authors and the physiological part measured could be correlated.
We included the following text to the manuscript:
Depression, anxiety, or stress lead to an increase in heart rates [3]. As a result, blood flow to the heart decreases and the body produces more cortisol. Over time, these effects can lead to heart disease. […] A study with cognitive behavioral therapy showed increases in heart rate variability in patients with Major Depression. Initially more depressed patients showed the most pronounced cardiovascular improvements through cognitive behavioral therapy. These findings provide new insights into the effects of psychological treatment against depression [10].
Round 2
Reviewer 2 Report
Substantial additions were made to the document; however, some points are still to be clarified.
I consider it essential to offer information, mainly differences about in the severity of the two pathologies studied; that is, the same benefit is obtained from the therapy in both pathologies? (this had already been asked in the first review).
Author Response
Dear Reviewer,
we highly appreciate your critical input to our manuscript entitled " Effects of music therapy on vital signs in children with chronic disease". Our changes to the original manuscript are highlighted in red again.
Substantial additions were made to the document; however, some points are still to be clarified.
I consider it essential to offer information, mainly differences about in the severity of the two pathologies studied; that is, the same benefit is obtained from the therapy in both pathologies?
We included the following text to the manuscript:
In terms of heart rate and oxygen saturation, patients benefit equally from the music therapy intervention, regardless of the severity of the disease. The heart rate (-17.9 beats per min SCU, -18.9 beats per min ICU) and oxygen saturation (2.4% SCU, 2.1% ICU) improved nearly independently whether the child is at SCU or at ICU. The changes in values heart rate and oxygen saturation are almost the same. But baseline and post-therapy values of heart rates were lower and of oxygen saturation were higher in children at SCU than at ICU, which can be explained by the severity of the disease [18]. Blood pressure decreased significant at SCU but not significant at ICU. This shows that the positive effects of music therapy of blood pressure decrease with increasing disease severity. All children were unsedated at the time of music therapy. In a future study, the mental state of the children should also be analyzed in addition to the vital signs before and after a music therapy session.
